# A Data-Driven System Identification Method for Random Eigenvalue Problem Using Synchrosqueezed Energy and Phase Portrait Analysis

**DOI:** 10.3390/s23073421

**Published:** 2023-03-24

**Authors:** Swarup Mahato, Arunasis Chakraborty, Paulius Griškevičius

**Affiliations:** 1Department of Mechanical Engineering, Kaunas University of Technology, 51424 Kaunas, Lithuania; paulius.griskevicius@ktu.lt; 2Department of Civil Engineering, Indian Institute of Technology Guwahati, Guwahati 781039, Assam, India; arunasis@iitg.ernet.in

**Keywords:** random eigenvalue, wavelet transform, synchrosqueezed transform, k-means clustering, asymptotic integral, modal identification

## Abstract

The primary purpose of this research is to evaluate the uncertainty associated with modal parameter estimation for an inverse dynamic problem in which the structural parameters are random. The random nature of the structure’s parameters will be reflected in the modal features of the respected system. However, this may result in additive/subtractive errors in modal parameter identification, affecting the identification technique’s efficiency. With this in mind, the present study aims to develop an automated modal identification algorithm for a random eigenvalue problem. This is achieved by a recently developed advanced version of the wavelet transform (i.e., synchrosqueezing), which offers better resolution. Using this technique, the measured responses are transformed into a time-frequency plane, which is further processed by unsupervised learning using K-means clustering for quantification of the modal parameters. This automated identification is repeated for an ensemble of measurements to quantify the random eigenvalues in a statistical sense. The proposed methodology is first tested using simulated time histories of a two degree-of-freedom (dof) system. It is followed by an experimental validation using a beam whose mass matrix is random. The numerical results presented in this work clearly demonstrate the performance (i.e., in terms of efficiency and accuracy) of the proposed output-only automated data-driven identification scheme for random eigenvalue problems.

## 1. Introduction

Uncertainty is a natural occurrence that scientists and engineers face on a regular basis. It is often simplified to a certain extent for simplicity of analysis and design. Nonetheless, it is unavoidable in other circumstances because simplifying dilutes the intent. To deal with the challenges in these settings, designers must use advanced modeling and analysis techniques. For instance, consider a dynamic system having uncertainty in its material and/or geometric properties. This leads to uncertain modal features, which pose significant mathematical hurdles to evaluate [1,2,3].

This problem can be addressed in the Bayesian framework [4], which, in principle, is a predictor-corrector algorithm based on observations. In this approach, mathematical modeling is started based on the initial guess, which is recursively updated using observations or measurements to obtain an optimal model of the physical system [5]. Here, it should be noted that the above iterative procedure demands observations, which in turn needs a physical system. Therefore, there is a constant need for forward and inverse modeling (i.e., quantification and identification) of the uncertainty associated with the eigenvalues. Reynders et al. [6] have tried to quantify the uncertainty associated with the modal parameters using the stochastic subspace identification (SSI) technique. SSI is an output-only state-space-based approach, which can extract modal information (i.e., modal frequencies, modal damping ratios, and mode shapes). In recent work, Reynders et al. [7] investigated the same topic in further depth, using response covariance to update the state-space model and extract the necessary information. In this process, different ensembles are used with different sensor placements; hence, the magnitude of error in observations varies in different sets. Thus, this study has reported different modeling errors, which are the outcome of different sensor placements based on algebraic schemes to estimate the output correlation sequence. It is followed by an eigen decomposition of the updated state-space model to extract the modal frequencies, mode shape, and damping ratios. The efficiency of SSI, like other system identification approaches, grows with the number of observations, although this methodology is prone to produce a significant number of spurious modes. A great amount of post-processing is required to remove these spurious modes. Researchers [8,9] select stringent validation criteria that include many levels of separation approaches such as blind clustering, hierarchical clustering, DBScan, mathematical mode reduction, and so on. Many SSI versions have been created to alleviate this burdensome post-processing. Among them, SSI-cov (i.e., covariance-driven stochastic subspace identification) [10,11,12] is very popular. These techniques are often merged with other methods with a small modification for practical application. For example, NExT (i.e., natural excitation technique) is combined with ERA (i.e., eigen realization algorithm) for operational modal analysis [12,13,14]. The reason behind this adaptation is to convert the forced vibration response into a free vibration prior to the extraction of modal features and hence, has multiple sources of model and estimation error leading to different levels of uncertainty. In this context, the eigen realization algorithm is based on Henkle’s matrix obtained from the free vibration response, which is performed by varying the order of the Henkle matrix while observing the convergence of the estimated eigenvalues, i.e., frequencies. As these SSI variants work on the covariance structure of the system responses, they offer standard deviation associated with the identified parameters, which are often used to quantify the uncertainty associated with them. Au [15,16] has evaluated the uncertainty associated with the modal parameters (i.e., frequencies, damping, and mode shapes) using a Bayesian approach. In another work, Au et al. [17,18] have modified their earlier proposal for a Bayesian approach to quantify modal uncertainty considering many factors, such as data recording, experimental configurations, signal-to-noise ratio, data length factors, among many others, and have prescribed guidelines for practical users. Ghiasi et al. [19] propose a nonprobabilistic method that combines wavelet packet decomposition and support vector algorithm. Wavelet packet decomposition extracts the energy features of the acceleration response, and then it is used in the support vector machine to represent a single damage scenario. After that, these scenarios are compared with the healthy state and the probability of matching is evaluated. This is called the probability of the existence of damage. In the true sense, this approach is unsuitable for modeling detection uncertainty as the final outcome largely depends on the wavelet transformation, i.e., for the same damage scenario, one can get a different value of the probability of damage existence by changing the level of decomposition. Considering recent algorithm developments, the authors/engineers [20,21,22] are attempting to model the uncertainty of detection using machine learning techniques. The machine learning or artificial neural network techniques are combined with the conventional operational modal analysis methods (i.e., principle component analysis, SSI, frequency response function, clustering, etc.) to extract features and evaluate the uncertainty associated with the identified parameters. However, this strategy’s effectiveness heavily relies on the correct application of conventional approaches. Mo et al. [23] present a polynomial-based model for detecting uncertainty in modal parameters and demonstrate its efficacy through experimental investigation. In another application, Niu et al. [24] study the effect of measurement uncertainty on the efficiency of parameter estimation. In contrast to the previous studies, this work [24] shows that a low signal-to-noise ratio is not always helpful for better parameter extraction.

The studies mentioned above mostly consider uncertainties associated with the experimental procedure and identification scheme, but not with the physical system. Thus, the problem becomes more complex, where the parent system exhibits uncertainty besides the above-mentioned sources. In this context, Rogers [25] and Rudisill [26] have proposed a mathematical framework for the forward problem of modeling uncertainty propagation through eigenvalues. Song et al. [27] have studied changes in random eigenvalues due to changes in random structural properties of a truss (i.e., with random cross-sections). In this formulation, Taylor’s series expansion of mass and stiffness matrices about their respective mean values is used. As the eigenvalues are obtained from Taylor’s series, the estimation of higher-order derivatives plays a major role in the uncertainty quantification associated with them. To simplify this process, Nair et al. [28] have proposed an approximation, which uses up to the first order in Taylor’s series. This perturbation-based approximation offers an improvement over the previous technique. Pradlwarter et al. [29] have developed an algorithm to reduce computational difficulty in calculating random eigenvalues and eigenvectors for large systems. In this work, instead of using general Monte Carlo simulation (MCS) to evaluate the random nature of the frequencies, a subspace iteration-based technique within MCS is used to save computational costs. Examples presented in their work show the efficiency of this technique to characterize the uncertainty associated with mass and stiffness parameters. Yana et al. [30] have developed an output-only methodology to quantify the uncertainty associated with the identified modal parameters. This proposal uses the random decrement technique to generate the free response time histories from the recorded ambient vibration. Once the free responses are obtained, Bootstrap sampling is used to generate ensembles with the same characteristics as the original response. These simulated responses are processed by wavelet transformation (WT) to estimate the modal parameters (i.e., frequencies and damping ratios). The probability distribution functions are made from the modal parameters, which dictate the amount of uncertainty present in the system. This work is a good example of an output-based identification technique that uses simulated samples to quantify uncertainty. The simulation (i.e., Bootstrap filtering) in this example has a major role in identifying uncertainty associated with the modal parameters in addition to signal processing (i.e., WT). Adhikari and Friswell [31] have proposed two different perturbation expansion-based methodologies (i.e., one is maximum entropy-based while another is an asymptotic approximation of integral) to estimate the probability distribution function (*pdf*) that describes the random eigenvalues. The advantages of these techniques are that they do not require the assumption of small randomness as in the earlier works and also do not impose any restriction on the type of uncertainty (i.e., Gaussian or non-Gaussian). The efficiency of both these methods is tested with different conditions and is found to be fairly accurate except for closely spaced frequencies. Adhikari [32] have also derived the joint moments of the random eigenvalues with the help of an asymptotic approximation of multidimensional integrals, for a linear random system. In addition to theoretical development, Adhikari et al. [33,34] have conducted laboratory experiments to demonstrate the effect of system uncertainty on the modal frequencies. For this purpose, a beam and a plate model are tested, where discrete masses are placed randomly in addition to the actual mass of the original system. The movable masses used in this study to simulate randomness constitute only 10% of the main structure. An ensemble of responses is generated using this set-up, which is processed through a Fast Fourier Transformation (FFT) analyzer to identify the modal frequencies, followed by statistical analysis to characterize its underlying randomness. Rahman [35] have developed a Fourier polynomial-based formulation to evaluate the uncertainty associated with the eigenfrequencies in a close form. The numerical studies reported in this paper show a close match with MCS provided the orthonormal polynomials used in this formulation are continuous. In another work, Rahman and Yadav [36] have also proposed a different approach using polynomial chaos expansion to evaluate random eigen frequencies in close-form. These studies are mostly focused on the forward problem, where an ensemble of eigenvalues or system responses is characterized by the predefined uncertainties associated with the structural parameters. In the recent past, researchers have tried to quantify the sensitivity of the modal parameters in the presence of randomness in the system [37,38]. In these approaches, the frequency response functions are extracted from the system responses along with the confidence levels and bounds.

Based on the preceding research, it is possible to conclude that the forward problem of modeling random eigenvalues has gotten a great deal of attention. Yet, finding these uncertainties through inverse analysis of the same problem has remained a challenge. With this in mind, the current research focuses on the development of a practical methodology for tracking and quantifying the random nature of the eigenvalues and associated modal parameters derived from measurement. To do so, the forward problem is first discussed, which forms the basis for the inverse identification. It is followed by the motivations/objectives of this work. Finally, the proposed automated identification strategy is elaborated along with its experimental verification.

## 2. Uncertainty Propagation through Eigen Value Problems

In this section, a brief overview of uncertainty propagation through eigenvalue problems often encountered in dynamical systems is presented with only a relevant equation to set the prelude of the present study. It mainly focuses on the modeling and associated problems when the parameters of a linear dynamic system have uncertainty. The dynamic equilibrium equation of a linear multi-degree of freedom system takes the form
(1)$$\left[M\right]\left\{\ddot{u}\left(t\right)\right\}+\left[C\right]\left\{\dot{u}\left(t\right)\right\}+\left[K\right]\left\{u(t)\right\}=\left\{P(t)\right\}$$
Here [M], [C] and [K] all $\in R^{J\times J}$ are the system matrices corresponding to mass, damping, and stiffness, respectively. The displacement vector is denoted by u and the upper dot represents the derivative with respect to time. In the above equation, P(t) is the generalized external force vector. The solution to this problem is straightforward for a system with constant coefficients (i.e., [M], [C] and [K]). In general, this coupled system is transformed into a set of independent modal coordinates, where it vibrates with its characteristic natural frequencies (i.e., $\omega_{n}$) and mode shapes (i.e., Φ). As the system parameters are deterministic, these natural frequencies and mode shapes are also deterministic in nature. However, for a system whose parameters (either material and/or geometric) are random, the modal parameters also become random, which needs to be characterized first. Let $x\in R^{n\times n}$ consist of Yong’s modulus, material properties, Poison’s ratio, membrane orientation, and geometric variable, e.g., length, width, and thickness. This randomness is bound to affect the mass and stiffness of the system and ultimately the natural frequencies. Therefore, *x* is the vector of random variables that represent system parameters (i.e., $x=x_{1},x_{2},...,x_{n}$), then the eigenvalue problem associated with Equation (Equation 1) can be represented in the following form
(2)[K(x)]Φ(x)=Λ(x)[M(x)]Φ(x)
In this equation, Λ(x) is the square of the natural frequencies, i.e., $\Lambda \left(x\right)=\omega_{n}^{2}\left(x\right)$. It may be noted that the mass and stiffness are assumed to be random while damping follows Raleigh’s proportionality model. Thus, the randomness associated with damping can be modeled using the properties of mass and stiffness. The uncertainties of the mass and stiffness of the system are propagated to the eigenvalues ($\Lambda_{i}$) and the eigen vectors ($\Phi_{i}$) through the relation described in Equation (Equation 2), where J≠n. The main question that arises here is what could be the distribution of these modal parameters? In other words, the main objective of the forward problem is to find the probability density function of Λ(x) and Φ(x). There are many books and literature available on this topic [27,29,39] with details of different formulations for the solution to this problem. These strategies can be broadly classified into two major approaches—(i) perturbation-based techniques, which are applicable for small randomness and (ii) asymptotic integral-based techniques, which are inherently tuned for more general applications. In the following subsections, these methods are briefly discussed with key equations to set the background for the present work on inverse identification.

### 2.1. Perturbation Approach

In this approach, mass, stiffness and modal parameters are expressed in Taylor’s series, where upto first order term is considered, i.e.,
(3a)$$\begin{array}{cc} \left[K\right] & =\left[K\left(\alpha \right)\right]+\sum_{k=1}^{n}{\left[K{\left(x\right)}_{,k}\right]}_{x_{k}=\alpha_{k}}\left(x_{k}-\alpha_{k}\right)+... \end{array}$$
(3b)$$\begin{array}{cc} \left[M\right] & =\left[M\left(\alpha \right)\right]+\sum_{k=1}^{n}{\left[M{\left(x\right)}_{,k}\right]}_{x_{k}=\alpha_{k}}\left(x_{k}-\alpha_{k}\right)+... \end{array}$$
(3c)$$\begin{array}{cc} {}[\Phi_{j}] & =\left[\Phi_{j}\left(\alpha \right)\right]+\sum_{k=1}^{n}{\left[\Phi_{j}{\left(x\right)}_{,k}\right]}_{x_{k}=\alpha_{k}}\left(x_{k}-\alpha_{k}\right)+... \end{array}$$
(3d)$$\begin{array}{cc} {}[\Lambda_{j}] & =\left[\Lambda_{j}\left(\alpha \right)\right]+\sum_{k=1}^{n}{\left[\Lambda_{j}{\left(x\right)}_{,k}\right]}_{x_{k}=\alpha_{k}}\left(x_{k}-\alpha_{k}\right)+... \end{array}$$
In the above equations, α is the reference point for the expansion. Now, to quantify the sensitivity of the eigenvalues and eigenvectors, Equation (Equation 2) is differentiated with respect to $x_{k}$, which leads to
(4)$$\begin{array}{cc} \left[K{\left(x\right)}_{,k}\right]\Phi_{j}\left(x\right)+\left[K\left(x\right)\right]\Phi {\left(x\right)}_{j,k}= & \\ \Lambda {\left(x\right)}_{j,k}\left[M\left(x\right)\right]\Phi {\left(x\right)}_{j}+\Lambda {\left(x\right)}_{j}\left[M{\left(x\right)}_{,k}\right]\Phi {\left(x\right)}_{j}+\Lambda {\left(x\right)}_{j}\left[M{\left(x\right)}_{,k}\right]\Phi {\left(x\right)}_{j,k} \end{array}$$
where subscript *j* represents mode number while subscript, k represents first derivative with respect to $x_{k}$ (i.e., $k^{th}$ random parameter). Taking expectation on both sides after substituting Equation (3) in Equation (Equation 4) and simplifying them lead to the following expression for stochastic sensitivity of the eigenvalues [27]
$$\begin{array}{cc} E[\Lambda_{j}{\left(x\right)}_{,k}] & =\chi [{\left\{\Phi_{j}\left(\alpha \right)\right\}}^{T}\left({\left[K_{k}\right]}_{x_{k}=\alpha_{k}}-\Lambda_{j}\left(\alpha \right){\left[M_{k}\right]}_{x_{k}=\alpha_{k}}\right)\left\{\Phi_{j}\left(\alpha \right)\right\} \end{array}$$
(5a)$$\begin{array}{cc} \; & -\sum_{p=1}^{m}\sum_{q=1}^{m}\Lambda_{j}\left(\alpha \right){\left\{\Phi_{j}\left(\alpha \right)\right\}}^{T}\left({\left[M_{k}\right]}_{x_{k}=\alpha_{k}}\left\{\Phi_{j}\left(\alpha \right)\right\}+{\left[M_{k}\right]}_{x_{k}=\alpha_{k}}\left\{\Phi_{j}\left(\alpha \right)\right\}\right)Cov\left(x_{p},x_{q}\right)] \end{array}$$
(5b)$$\begin{array}{cc} \chi & =1+\sum_{p=1}^{m}\sum_{q=1}^{m}{\left\{\Phi_{j}\left(\alpha \right)\right\}}^{T}{\left[M{\left(x\right)}_{,k}\right]}_{x_{k}=\alpha_{k}}\left\{\Phi_{j}\left(\alpha \right)\right\}Cov\left(x_{p},x_{q}\right) \end{array}$$
Similar to the eigenvalues, the stochastic sensitivity of the eigen vectors can be quantified by the following expression [27]
(6)$$E\left[\Phi_{j}{\left(x\right)}_{,k}\right]=\sum_{r=1}^{n}b_{jkr}{\left\{\Phi_{r}\right\}}_{\alpha }$$
The expression for the coefficients $b_{ikr}$ in the above equation is omitted here to avoid repetition. This formulation for first-order sensitivity of the eigenvalues and vectors considers the mean of the random variables as the reference point, which is applicable for problems having small randomness. Moreover, its performance is limited to random variables following normal distributions only.

To address these issues, Adhikari and Frieswell [31] have proposed new formulations, where up to second order term in Taylor’s series is considered. Besides additional terms, the optimal points are used for *pdf* estimation, instead of mean values. Thus, the eigen function Λ(x) is expressed in Taylor’s series, as follows
(7)$$\Lambda_{j}\left(x\right)=\sum_{\left|n\right|\geq 0}\frac{{\left(x-\mathrm{ff}\right)}^{n}}{n!}\left(\partial^{n}\Lambda_{j}\right)\left(\mathrm{ff}\right)$$
In above equation, $\partial^{n}\left(.\right)$ represents $n^{th}$ order partial differential of the function. So, n=1 corresponds to the gradient vector while n=2 provides the Hessian matrix evaluated at the reference point α. As mentioned above, this formulation uses optimal points {ff} in the sample space instead of mean and the *pdf* of the eigenvalues are obtained from the associated moment-generating function. To quantify the statistical properties of the random vector *x*, the probability density function is expressed using the likelihood function as follows
(8)$$q_{x}\left(x\right)=exp\left\{-L\left(x\right)\right\}$$
where −L(x) is the log-likelihood function, which takes the following form for *n*-dimensional multivariate Gaussian vector with a mean μ∈R and covariance Σ∈R, i.e.,
(9)$$L\left(x\right)=\frac{n}{2}\ln\left(2\pi \right)+\frac{1}{2}\ln∥\Sigma ∥+\frac{1}{2}{\left(x-\alpha \right)}^{T}\Sigma^{-1}\left(x-\alpha \right)$$
Using this definition of *pdf*, the $r^{th}$ moment of the eigenvalues are defined as follows
(10)$$E\left[\Lambda_{j}^{r}\left(x\right)\right]=\int_{R^{n}}\Lambda_{j}^{r}\left(x\right)q_{x}\left(x\right)dx=\int_{R^{n}}\Lambda_{j}^{r}\left(x\right)exp\left\{-L\left(x\right)\right\}dx=\int_{R^{n}}exp\left[L\left(x\right)-r\{ln\Lambda_{j}\left(x\right)\}\right]dx$$
The above expression for multi-dimensional integral is difficult to obtain (although not impossible numerically). However, Adhikari and Frieswell [31] have proposed a saddle point approximation of this integral, which is mostly governed by the global minima of the function $\left\{L\left(x\right)-ln\Lambda_{j}\left(x\right)\right\}$ within the sample space of x, which leads to the relation $\Lambda_{j}\left(x\right)\partial L\left(x\right)=\partial \Lambda_{j}\left(x\right)$. This expression can be solved in an iterative framework to obtain the optimal points for any given distributions of x. A closed-form solution for Gaussian distribution can be obtained as follows
(11)$$\alpha =\mu +\frac{1}{\Lambda_{j}\left(\alpha \right)}\Sigma \partial \Lambda_{j}\left(x\right){|}_{\alpha }$$
Once the optimal points are evaluated, moment generating function is adopted to evaluate the moments of the random eigenvalues, which has the form $M_{\Lambda_{j}}\left(s\right)=E\left[exp\left\{s\Lambda_{j}\left(x\right)\right\}\right]$. Finally, the $r^{th}$ order moment can be obtained from the moment-generating function using the following expression
(12)$$E\left[\Lambda_{j}^{r}\left(x\right)\right]=\frac{d^{r}}{ds^{r}}lnM_{\Lambda_{j}}{\left(s\right)|}_{s=0}$$
The close form expression for the first four moments are derived by Adhikari and Friswell [31].

### 2.2. Asymptotic Integral Approach

Equation (Equation 12) in the above sub-section provides the moments using the moment-generating function at the optimal points. However, multi-dimensional integral in Equation (Equation 10) can be evaluated using asymptotic integral [31], where the major contribution originates from the optimal points α obtained by minimizing L in Equation (Equation 10). Using the optimal condition, it can be shown that the asymptotic approximation of the $r^{th}$ moment of the random eigenvalues takes the following form
(13)$$E\left[\Lambda_{j}^{r}\left(x\right)\right]\approx {\left(2\pi \right)}^{n/2}\Lambda_{j}^{r}\left(\alpha \right)exp\left\{-L\left(\alpha \right)\right\}{∥D_{L}\left(\alpha \right)+\frac{1}{r}d_{L}\left(\alpha \right)d_{L}{\left(\alpha \right)}^{T}-\frac{r}{\Lambda_{j}\left(\alpha \right)}D_{\Lambda_{j}}\left(\alpha \right)∥}^{1/2}$$
Once the moments are derived using the above equation or as in Equation (Equation 12), *pdf* of the eigenvalues, i.e., $p_{\Lambda_{j}}\left(x\right)$ can be evaluated using Maximum Entropy Method. This technique ideally solves an optimization problem such that $r^{th}$ moment obtained from the above expression match with the same obtained from $p_{\Lambda_{j}}\left(x\right)$, i.e.
(14)$$E\left[\Lambda_{j}^{r}\left(x\right)\right]=\int_{0}^{\infty }x^{r}p_{\Lambda_{j}}\left(x\right)dx,\;r=1,2,3,...$$
This is solved using the Lagrange multiplier approach and the calculus of variations. If only first two moments are used in Equation (Equation 14), the close form solution of the *pdf* takes the following form of the truncated Gaussian density function
(15)$$p_{\Lambda_{j}}\left(x\right)=\frac{1}{\sqrt{2\pi }\varphi \left({\hat{\Lambda }}_{j}/\sigma_{j}\right)}exp\left\{-\frac{{\left(x-{\hat{\Lambda }}_{j}\right)}^{2}}{2\sigma_{j}^{2}}\right\}\;...\;x\geq 0$$
In the above equation, $\sigma_{j}$ is the square root of the difference between the second moments and square of the mean of the eigenvalues (i.e., ${\hat{\Lambda }}_{j}^{2}$).

### 2.3. Objectives for the Inverse Data-Driven Identification

The above discussion demonstrates the nature of the eigenvalues when the fundamental structural parameters are random. It outlines the forward problems solved by the previous researchers, which are often encountered in engineering applications. In this context, its identification poses an open problem to the engineers and researchers, which is addressed in this paper. It involves precise quantification of the *pdf* that governs the nature of these eigenvalues. Thus, the objectives of the proposed inverse problem are as follows

Develop an efficient output-only signal processing tool that can identify the modal parameters from the measured acceleration responses without any user intervention. The reason behind this approach is to automate the process, where a large number of tests are necessary for handling stochastic structural systems. This is achieved by synchrosqueezed transform in this work, which offers improved resolution in the time-frequency domain compared to wavelet-based signal processing.Since a single measurement is an outcome of a random process, it is proposed to be repeated for an ensemble and the extraction of the modal features for the complete set is automated with the help of a clustering algorithm. In this context, k-means clustering is adopted for unsupervised learning of the spectrogram obtained from synchrosqueezed transformation, which reflects arrange the modal energy in the time-frequency plane. This step is as important as the accurate estimation of modal parameters in the previous step, as it helps to manage large data produced from repeated trials.Once this data-driven process is repeated for the ensemble, the underlying randomness of the modal parameters is quantified by the *pdf* of these parameters, which is experimentally verified to study the efficiency and accuracy of the proposed identification strategy for the random eigenvalues.

The details of the proposed inverse approach are discussed in the following sections.

## 3. Inverse Analysis of Random Eigen Values

In this section, the proposed automated data-driven strategy for inverse identification of the random eigenvalues is discussed in detail. As described in the objectives, this problem is addressed with the help of an advanced version of wavelet-based time-frequency analysis along with the data clustering to automate the whole process, which is followed by the quantification of the underlying *pdfs*.

### 3.1. Continuous Wavelet Transform & Synchrosqueezing

Continuous wavelet transform is a powerful signal processing tool that helps to extract the time-localized frequency details. The theory is well developed and the reader may refer [40,41] for the details of this integral transform. In this section, a brief overview of continuous wavelet transform is presented, which forms the backbone of the proposed data-driven identification scheme.

As the name suggests, wavelets are a collection of localized and dilated versions of a function Ψ(t) that convolutes with a time function $f\left(t\right)\in L^{2}\left(R\right)$ to convert it into a two-dimensional sequence, i.e.,
(16)$$W_{\psi }f\left(a,b\right)=\frac{1}{\sqrt{\left|a\right|}}\int_{-\infty }^{+\infty }f\left(t\right)\Psi^{*}\left(\frac{t-b}{a}\right)dt$$
In this process, Ψ(t) must satisfy two important criteria—(i) it must have finite energy (i.e., $\int_{-\infty }^{+\infty }\left|\Psi^{2}\left(t\right)\right|dt<\infty$) and (ii) it must comply with the admissibility criteria (i.e., $C_{\Psi }=\int_{0}^{\infty }\frac{{\left|\Psi \left(\omega \right)\right|}^{2}}{\left|\omega \right|}d\omega <\infty$). The parameters *a* and *b* in $\Psi_{a,b}\left(t\right)=\Psi \left(\frac{t-b}{a}\right)$ correspond to the scale and time localization of the wavelet coefficients $W_{\Psi }f\left(a,b\right)$, while ${\left(\right)}^{*}$ represents a complex conjugate. On inverse transforming Equation (Equation 16), the original time signal can be traced back from its wavelet coefficients using the following expression
(17)$$f\left(t\right)=\frac{1}{C_{\Psi }}\int_{-\infty }^{+\infty }\int_{0}^{+\infty }\frac{1}{a^{2}}W_{\Psi }f\left(a,b\right)\Psi_{a,b}\left(t\right)dadb$$
The convolution integral in time domain (i.e., Equation (Equation 16)) is slow and computationally expensive. It can be made faster by transforming it into a frequency domain using Fast Fourier Transform. This is done by Fourier transform of f(t) (i.e., $F\left(\omega \right)=\int_{-\infty }^{+\infty }f\left(t\right)exp\left(i\omega t\right)d\omega$) in Equation (Equation 16), which leads to
(18)$$W_{\psi }f\left(a,b\right)=\sqrt{2\pi a}\int_{-\infty }^{+\infty }F\left(\omega \right)\Psi^{*}\left(a\omega \right)\exp\left(i\omega b\right)d\omega$$
The wavelet coefficients in Equation (Equation 16) (or Equation (Equation 18)) is useful to study the relative distribution of signal energy in different scales and time, i.e., the energy density in two dimensions $E_{f(t)}={\left|W_{\Psi }f\left(a,b\right)\right|}^{2}$ a.k.a scalogram. This feature will be utilized in the proposed automated uncertainty quantification associated with the random eigenvalues. The scalogram also ensures equality of energy in different domains, i.e.,
(19)$$E_{f(t)}=\left||f\left(t\right)|\right|=\int_{-\infty }^{+\infty }{\left|f\left(t\right)\right|}^{2}dt=\frac{1}{C_{\Psi }}\int_{-\infty }^{+\infty }\int_{0}^{+\infty }{\left|W_{\Psi }f\left(a,b\right)\right|}^{2}dadb$$
Unlike Fourier transform, where the exponential kernel is used, continuous wavelet transform adopts different basis functions, i.e., Ψ(t) for different purposes, e.g., Haar, Mexican Hat, Morlet, Morse [41,42,43] etc. Among them, complex Morlet is widely used in different applications and is adopted in this study. This basis function has the following form
(20)$$\begin{array}{cc} \Psi (t) & =\pi^{-1/4}\left\{e^{i\omega_{c}t}-e^{-\omega_{c}^{2}/2}\right\}e^{-t^{2}/2}\\ \; & \approx \pi^{-1/4}e^{i\omega_{c}t}e^{-t^{2}/2}\;...\;\omega_{c}>>0 \end{array}$$
where $\omega_{c}$ is the central frequency of this analytic basis function, which is a complex sinusoid (i.e., $exp(i\omega_{c}t)$) modulated by the Gaussian window (i.e., $exp(-t^{2}/2)$). The value of the central frequency is 5rad/s unless otherwise specified. The frequency signature of this basis function has the following form
(21)$$\Psi \left(\omega \right)=2^{1/2}\pi^{1/4}e^{-{\left(\omega -\omega_{c}\right)}^{2}/2}$$
The characteristic frequency ω is related to the scale *a* through the relation $\omega =\omega_{c}/a$. In this framework of the continuous wavelet transform, both *a* and *b* are continuous, where the second parameter is always positive. For numerical implementation, these parameters are often discretized (not discrete wavelet transform) such that $a_{j}=\sigma^{j}$ and $b_{i}=\left(i-1\right)\Delta b$, where both σ and Δb are constants. This discretization scheme of wavelet parameters leads to $\Delta a_{j}=\frac{\sigma^{2}-1}{2\sigma }a_{j}$, which ultimately helps in expressing the inverse wavelet transform (i.e., Equation (Equation 17)) in the following numerical form
(22)$$f\left(t\right)=\sum_{j=1}^{n_{b}}\sum_{i=1}^{n_{a}}\frac{K\Delta b}{a_{j}}W_{\Psi }f\left(a_{j},b_{i}\right)\Psi \left(\frac{t-b_{i}}{a_{j}}\right)$$
where $n_{a}$ and $n_{b}$ are the numbers of discrete points of scale and time parameters while $K={\left(4\pi C_{\Psi }\sigma \right)}^{-1}\left(\sigma^{2}-1\right)$ is a constant.

Although the above-mentioned continuous wavelet transform is, by definition, capable of extracting time-localized features of any signal, its scalogram often shows poor energy localization over a range of dilation parameters $a_{j}$. In this context, the instantaneous amplitude and phase of the two-dimensional wavelet coefficient array are defined as—$|W_{\Psi }f\left(a_{j},b_{i}\right)|$ and $\vartheta \left(a_{j},b_{i}\right)=tan^{-1}\frac{I(W_{\Psi }f\left(a_{j},b_{i}\right))}{R(W_{\Psi }f\left(a_{j},b_{i}\right))}$, where I(.) and R(.) represent the imaginary and real part of the analytic signal. For a real signal $f\left(t\right)\in L^{2}\left(R\right)$, whose wavelet coefficients are obtained using analytic basis function, the differential of $arg(W_{\Psi }f\left(a_{j},b_{i}\right))$ with respect to its scale vanishes on the ridge, i.e., $\frac{\partial \vartheta (a,b)}{\partial a}=0$. Using this property of the continuous wavelet transform, the ridge can be identified [44,45,46], which helps to quantify the energy localization in the scalogram. However, as the analytic basis function is modulated by a Gaussian window, its frequency signature does not decay in the vicinity of the ridge. In reality, it is spread over a region depending upon the effective width of the Gaussian window, no matter how small that may be. Due to this reason, a resolution associated with energy localization becomes poor, which ultimately affects the identification and often demands heuristic user intervention. This will be further elaborated with the help of numerical examples.

To address this issue, a reallocation or reassignment algorithm has been developed in the recent past [47,48,49], which offers an improved resolution, i.e., better localization characteristics. This reassignment (also known as synchrosqueezing) helps to decompose the signal into components that are well separated in frequency content and a direct summation of all these components brings back the original time function. This reallocation/reassignment operation has two steps—(i) identify the instantaneous frequency and (ii) evaluate synchrosqueezed coefficients in the vicinity of that instantaneous frequency. For the wavelet coefficients $W_{\Psi }f\left(a,b\right)$, the instantaneous frequency is obtained from the following expression
(23)$$\omega_{in}\left(a,b\right)=-i{\left[W_{\Psi }f\left(a,b\right)\right]}^{-1}\frac{\partial }{\partial b}W_{\Psi }f\left(a,b\right)$$
This operation helps to map the coefficients in the time-scale domain to the time-frequency domain (i.e., $\left(a,b\right)\to \left(\omega_{in},b\right)$) over a frequency width of Δω. In this process of synchrosqueezing, the wavelet coefficients within a bin $\Delta a_{k}=a_{k}-a_{k-1}$ are grouped together using the following relation
(24)$$S_{\Psi }f\left(\omega_{in},b\right)={\left(\Delta \omega \right)}^{-1}\sum_{a_{k}:\left|\omega \left(a_{k},b\right)-\omega_{in}\right|\leq \Delta \omega /2}W_{\Psi }f\left(a,b\right)a_{k}^{-3/2}\Delta a_{k}$$
With the help of these reassigned coefficients, the original signal can be reconstructed by the expression given below
(25)$$f\left(t=b\right)=R\left[C_{\psi }^{-1}\sum_{l}S_{\Psi }f\left(\omega_{l}=\omega_{in},b\right)\Delta \omega \right]$$
As an analytic basis function is used in this work, the synchrosqueezed coefficients also have real and imaginary components, i.e., $S_{\Psi }f\left(\omega ,b\right)=\left|S_{\Psi }f\left(\omega ,b\right)\right|exp\left(i\psi \left(a,b\right)\right)$, where |.| represents the modulus and ψ(.) is the argument. These two pieces of information with the superior resolution are proposed to be utilized for the automated identification of random eigenvalue problems, which are described below.

### 3.2. Proposed Energy and Phase Portrait Analysis for Inverse Random Eigen Value Problem

Beginning with this problem, let us consider the governing equation of motion, i.e., Equation (Equation 1). On wavelet transforming both sides of this equation, the following version of the dynamic equilibrium is obtained in the time-frequency plane
(26)$$\left[M\right]\frac{\partial^{2}}{\partial b^{2}}W_{\Psi }u\left(a_{j},b\right)+\left[C\right]\frac{\partial }{\partial b}W_{\Psi }u\left(a_{j},b\right)+\left[K\right]W_{\Psi }u\left(a_{j},b\right)=W_{\Psi }P\left(a_{j},b\right)$$
Similarly, the initial conditions can be transformed into the new domain for the complete description of the system. The remarkable feature of this equation is that the basic structure of the dynamic equilibrium is retained in the wavelet domain. Thus, the coupling in the original domain manifested through the constant coefficient matrices is also carried forward in this new domain. Due to this reason, the modal transformation used in the original domain also holds in the time-frequency plane, which can be expressed in the following form
(27)$$\begin{array}{cc} \left\{u\right\} & =[\Phi ]\{Z\}\\ \Rightarrow \left\{W_{\Psi }u\left(a_{j},b\right)\right\} & =\left[\Phi \right]\left\{W_{\Psi }Z\left(a_{j},b\right)\right\} \end{array}$$
Equation (Equation 27) clearly shows that the modal matrix, i.e., Φ is unchanged for the linear time-invariant system after transformation. Using this relation in the transformed domain, the coupled system can be expressed in the modal coordinates for every scale parameter $a_{j}$. Substituting modal transformation for any $a_{j}$ and *b* (i.e., Equation (Equation 27) in Equation (Equation 26)), the modal governing equation in wavelet domain takes the following form
(28)$$\begin{array}{cc} \left[M\right]\left[\Phi \right]\frac{\partial^{2}}{\partial b^{2}}\left\{W_{\Psi }Z\left(a_{j},b\right)\right\}+\left[C\right]\left[\Phi \right]\frac{\partial }{\partial b}\left\{W_{\Psi }Z\left(a_{j},b\right)\right\}+\left[K\right]\left[\Phi \right]\left\{W_{\Psi }Z\left(a_{j},b\right)\right\} & =\{W_{\Psi }P\left(a_{j},b\right)\}\\ \Rightarrow \frac{\partial^{2}}{\partial b^{2}}W_{\Psi }z_{n}\left(a_{j},b\right)+2\eta_{n}\omega_{n}\frac{\partial }{\partial b}W_{\Psi }z_{n}\left(a_{j},b\right)+\omega_{n}^{2}W_{\Psi }z_{n}\left(a_{j},b\right) & =W_{\Psi }{\bar{p}}_{n}\left(a_{j},b\right) \end{array}$$
In the above equation, $z_{n}$ is the $n^{th}$ modal displacement with associated frequency and damping ratio of $\omega_{n}$ and $\eta_{n}$, respectively. The right-hand side of the above equation has the $n^{th}$ modal load (i.e., $W_{\Psi }{\bar{p}}_{n}\left(a_{j},b\right)$) in the time-frequency plane. Since the dynamic characteristics remain unaltered in the modal domain, the convolution integral can be invoked to solve for a response. Thus, the modal response corresponding to a scale $a_{j}$ can be obtained as follows
(29)$$\begin{array}{cc} W_{\Psi }z_{n}\left(a_{j},b\right) & =\int_{0}^{b}h_{n}\left(b-\tau \right)W_{\Psi }{\bar{p}}_{n}\left(a_{j},\tau \right)d\tau \\ \; & =\sqrt{2\pi a_{j}}\int_{-\infty }^{+\infty }{\bar{p}}_{n}\left(\omega \right)\Psi^{*}\left(a_{j}\omega \right)exp\left(i\omega b\right)H_{n}\left(\omega \right)d\omega \end{array}$$
Here, $h_{n}\left(t\right)$ is the modal frequency response function. Together with $H_{n}\left(\omega \right)$, it forms the Fourier transform pair that dictates the modal characteristics of the parent system. In this context, it is worth mentioning that both $\omega_{n}$ and $\eta_{n}$ are random in nature in this study, which is aimed to be identified from the measured noisy responses of the system.

Without loss of generality, the wavelet basis function can be invoked at this stage, which is a complex Morlet in this study. Using its definition in Equation (Equation 21) and the expression for H(ω), the modal acceleration response in the wavelet domain due to an impulse can be expressed as follows
(30)$$W_{\Psi }{\ddot{z}}_{n}\left(a_{j},b\right)=2\pi^{3/4}\sqrt{a_{j}}\int_{-\infty }^{+\infty }\frac{-\omega^{2}e^{-{\left(a_{j}\omega -\omega_{c}\right)}^{2}/2}\left[cos\left(\omega b\right)+i.sin\left(\omega b\right)\right]}{\omega^{2}-\omega_{n}^{2}+i2\eta_{n}\omega_{n}\omega }d\omega$$
Now, the acceleration response (i.e., $\ddot{u}\left(t\right)$) in the wavelet domain can be obtained using $W_{\Psi }{\ddot{z}}_{n}\left(a_{j},b\right)$, which is used further for the identification of random eigenvalues in this study. The rationale behind the use of acceleration response is due to the ease of measuring this quantity accurately during any experiment as compared to displacement. Equation (Equation 30) for different $a_{j}$ and *b* forms the scalogram of the modal acceleration response of the system. Using these coefficients, the scalogram of the acceleration response ${\ddot{u}}_{\kappa }\left(t\right)$ (i.e., along the $\kappa^{th}$ dof) can be obtained, which has the following mathematical form
(31)$$W_{\Psi }{\ddot{u}}_{\kappa }\left(a_{j},b\right)=\sum_{n}\Phi_{\kappa ,n}W_{\Psi }{\ddot{z}}_{n}\left(a_{j},b\right)$$
Equation (Equation 31) establishes the fact that the energy of the response is localized in scales that correspond to the modal frequencies. Thus, the energy spectrum can reveal the location of modal frequencies in the spectrum. However, as mentioned earlier, the resolution of the above scalogram is not very sharp due to the decaying character of both complex Morlet wavelet Ψ(ω) and frequency response function $H_{n}\left(\omega \right)$ in the vicinity of its ridge. In other words, the scalogram offered by Equation (Equation 31) provides a weak portrait of the measured response as the mother wavelet is not completely localized in the frequency domain. This involves user intermittency to detect the ridge for modal identification [50,51].

To address this issue, synchrosqueeze transform is proposed in this study, which operates over the wavelet coefficients obtained from Equation (Equation 31). This transformation improves the resolution of its scalogram and hence, it helps to identify the energy localization precisely, which will be demonstrated further in the numerical analysis. The synchrosqueezed version of the modal acceleration can be obtained from Equation (Equation 30) following its definition in Equation (Equation 24), i.e.,
(32)$$S_{\Psi }{\ddot{z}}_{n}\left(\omega_{in},b\right)={\left(\Delta \omega \right)}^{-1}\sum_{a_{k}:\left|\omega \left(a_{k},b\right)-\omega_{in}\right|\leq \Delta \omega /2}W_{\Psi }{\ddot{z}}_{n}\left(a,b\right)a_{k}^{-3/2}\Delta a_{k}$$
Thus, the synchrosqueezed version of the $\kappa^{th}$ acceleration response has the following form
(33)$$S_{\Psi }{\ddot{u}}_{\kappa }\left(a_{j},b\right)=\sum_{n}\Phi_{\kappa ,n}S_{\Psi }{\ddot{z}}_{n}\left(a_{j},b\right)=\left|S_{\Psi }{\ddot{u}}_{\kappa }\left(\omega_{in},b\right)\right|exp\left(i\phi_{\kappa }\left(\omega_{in},b\right)\right)$$
where $\vartheta_{\kappa }\left(.\right)$ in the above equation is the phase angle. The scalogram obtained from this equation has better resolution compared to that obtained from wavelet coefficients. This is due to the reallocation of the coefficients within the bin, i.e., Δω. These improved coefficients are further utilized for mode localization and subsequent identification.

To demonstrate it further, let us consider two measurements (now onward will be referred to as channels) for the same structure, say ${\ddot{u}}_{\kappa }\left(t\right)$ and ${\ddot{u}}_{\iota }\left(t\right)$. From these two channels, respective synchrosqueezed signals are obtained as described in Equation (Equation 33), which are abbreviated by $S_{\kappa }$ and $S_{\iota }$. These signals are used to define the synchrosqueezed correlation, having a time delay of τ, which is expressed by
(34)$$R_{\kappa \iota }\left(a,\tau \right)=\frac{|\int S_{\kappa }^{*}\left(a,b\right).S_{\iota }\left(a,b-\tau \right)db|}{\left[\int \right|S_{\kappa }{\left(a,b\right)|}^{2}db\int |S_{\iota }\left(a,b\right){{|}^{2}db]}^{1/2}}$$
Here, it may be noted that sign of τ (i.e., + or −) depends upon the shift in Equation (Equation 34) relative to one another, which can be used to obtain scale dependent correlation for τ=0, ranging between 0 and 1. This, in other words, ensures that two channels will converge to the same mode when the correlation is 1. The above equation also reveals that the time localized version for τ=0 offers the normalized cross-energy spectrum with improved resolution as follows
(35)$$E_{S_{\kappa \iota }}\left(a,b\right)=\frac{|S_{\kappa }^{*}\left(a,b\right)\cdot S_{\iota }{\left(a,b\right)|}^{2}}{|S_{\kappa }{\left(a,b\right)|}^{2}{\left|S_{\iota }\left(a,b\right)\right|}^{2}}$$
These the reassigned normalized energy spectrum is proposed to be utilized for mode localization.

In this context, it is worth mentioning that the time-frequency spectrum often has spurious modes resulting from numerical analysis or input force. Although the impulse is used in this analysis, which negates the presence of frequencies corresponding to input force, the presence of spurious mode due convolution of wavelets with frequency response function can not be ruled out. Thus, its identification and segregation from modal frequencies play a key role in the automated inverse problem. This is addressed using two reallocated signals, whose local phase difference can be obtained as follows
(36)$$\Delta \vartheta_{\kappa \iota }\left(a,b\right)=\vartheta_{\kappa }\left(a,b\right)-\vartheta_{\iota }\left(a,b\right)$$
The above mentioned phase difference is in reality the phase angle of the cross synchrosqueezed transform, which has the following definition
(37)$$\begin{array}{cc} S_{\kappa \iota }\left(a,b\right) & =S_{\kappa }^{*}\left(a,b\right)\cdot S_{\iota }\left(a,b\right)\\ \; & =|S_{\kappa }\left(a,b\right)||S_{\iota }\left(a,b\right)|exp\left\{i\left(\vartheta_{\kappa }\left(a,b\right)-\vartheta_{\iota }\left(a,b\right)\right)\right\} \end{array}$$
As *a* (i.e., scale) changes, channels κ and ι offer different instantaneous phase angles whose difference vanishes for scale corresponding to a particular mode. The reason behind this vanishing nature is due to the fundamental property of modal vibrations when all dofs attain unison, i.e., maxima or minima or zero crossing. If the phase angle $\vartheta_{\kappa }$ and $\vartheta_{\iota }$ correspond to the same mode, their difference becomes 0 or π as the *dof*s are in unison. Thus, to verify the scale corresponding to modal frequency, the phase synchronization index (PSI) is proposed in this study, which takes the following form
(38)$$PSI\left(a\right)={\langle \sin\left(\Delta \vartheta_{\kappa \iota }\left(a,b\right)\right)\rangle }^{2}+{\langle \cos\left(\Delta \vartheta_{\kappa \iota }\left(a,b\right)\right)\rangle }^{2}$$
In the above equation, 〈.〉 represents the temporal averaging of the signal, whose PSI lies between 0 and 1. For two signals whose phase angles are invariant (i.e., mode), PSI becomes 1. This, in turn, reveals that the synchrosqueezed signals obtained from two different channels in the same mode must have both correlation and PSI values unity. Here, it is worth mentioning that two different channels are used only for demonstration and is not the prerequisite for this strategy. For κ=ι, i.e., same channel, auto-correlation value can be used for identification.

Once the modal frequencies are identified, mode-shapes and modal damping ratio can be estimated from the signals obtained from the inverse synchrosqueezed transform of coefficients in Equation (Equation 33), i.e., modal responses. Considering two channels, the ratio of the mode-shapes can be obtained as follows
(39)$$\Pi_{r}^{s}=\frac{\Phi_{r}^{s}}{\Phi_{r}^{1}}=\frac{{\ddot{u}}_{r}^{s}}{{\ddot{u}}_{r}^{1}}$$
However, identifying mode-shapes in this fashion has two major drawbacks—(i) at least two channels are necessary to take the ratio and (ii) the limited number of sensors for all practical cases can only provide a partial image of the global mode shape. This problem can be addressed by model updating [52,53], where the error between the natural frequencies obtained from the model and system identification scheme is minimized as follows
(40)$$\begin{array}{ccc} J & = & ∥W(\Lambda_{FE}-\Lambda_{SI})∥\\ \; & = & \sum_{i=1}^{l}\sum_{k=1}^{n}{\left[\sum_{r=1}^{l}W_{ir}\left({\Lambda_{FE}}_{rk}-{\Lambda_{SI}}_{rk}\right)\right]}^{2} \end{array}$$
subject to the following constrained conditions
(41a)$$\begin{array}{cc} \; & \Phi^{T}M\Phi =I \end{array}$$
(41b)$$\begin{array}{cc} \; & K^{T}=K \end{array}$$
(41c)$$\begin{array}{cc} \; & K\Phi =M\Phi \Lambda \end{array}$$
Here, $W_{ir}$ is the weight factor and Λ is the square of the natural frequencies while subscripts FE and SI correspond to the finite element model and system identification, respectively. Finally, the modal damping ratio is estimated from the slope of the logarithm of amplitude ($ln(|S_{\Psi }{\ddot{u}}_{\kappa }\left(\omega_{in},b\right)|)$) as it represents the free modal response [54,55].

### 3.3. Proposed Automated Data-Driven Identification

Above subsection describes the proposed synchrosqueezed wavelet transform-based modal parameter estimation. This description is self-sufficient for modal identification of a deterministic linear time-invariant problem as it is capable of estimating natural frequencies, mode shapes and modal damping ratio. This is carried out with the help of a scale-dependent energy spectrum and phase portrait. The process is repeated for a stochastic case, where the identification procedure needs to be adopted for a large number of tests without any user intervention to quantify the underlying model that describes the uncertainty associated with the modal parameters. For this purpose, k-means clustering is adopted in this study, which has the capability to automate the process and avoid user intermittency and associated error. The rationale behind the selection of this unsupervised scheme lies with synchrosqueezing, which helps to segregate the energy content of the signal in the time-frequency plane and hence, negates any additional supervision during clustering.

The k-means clustering primarily classifies a *n*-dimensional data set $\bar{X}={\bar{x}}_{i};$$i=1,2,...,N_{x}$ into *K* cluster $C=C_{k};$k=1,2,...,K, where each data point belongs to a specific cluster. It is a vector quantization process in signal processing that results in different Voronoi cells obtained by minimizing the intra-cluster variance $\sigma_{C_{k}}^{2}$. In this process, the squared Euclidean norm between the cluster mean and the points within that cluster is minimized. Thus, the objective function for this minimization is given by the following form
(42)$$J\left(\bar{x},v\right)=\underset{C}{\arg\min}\sum_{k=1}^{K}\sum_{x_{i}\in c_{k}}\|{\bar{x}}_{i}-\mu_{k}{\|}^{2}=\underset{C}{\arg\min}\sum_{k=1}^{K}\left|C_{k}\right|\sigma_{C_{k}}^{2}$$
The partitioning of the data initiates with *K* clusters having initial mean $\mu_{k};k=1,2,...,K$ and iteratively updates the boundary of the Voronoi cells in two major steps as follows

Assignment: Each data point is assigned to the appropriate Vononoi cell based on the nearest cluster mean $\mu_{k}$
(43)$$C_{k}^{i}=\left\{\bar{x}:\|{\bar{x}}_{p}-\mu_{k}^{i}{\|}^{2}\leq {\left\|{\bar{x}}_{p}-\mu_{j}^{i}\right\|}^{2}\forall j,1\leq j\leq k\right\}$$Update: Once the assignment is completed, the cluster means are updated as follows
(44)$$\mu_{k}^{i+1}=\frac{1}{C_{k}^{1}}\sum_{{\bar{x}}_{j}\in C_{k}^{i}}{\bar{x}}_{j}$$
The above iteration stops once the convergence is achieved. Readers may refer to [56,57] for the details of this algorithm and the convergence studies for efficient partitioning that uses the squared Euclidean norm instead of the regular norm. As for the present study, the synchrosqueezed scale-dependent energy spectrum is used as a data set for clustering, i.e., ${\bar{x}}_{i}=S_{n_{ij}}\left(a,b\right)$.

Once the data set is clustered, the contribution of each identified cluster can be estimated using the weight index as follows
(45)$$\lambda =\frac{\sum_{i=1}^{n}{\left|S_{ij}\left(a,b_{i}\right)\right|}^{2}}{\sum_{k=1}^{c}\sum_{\begin{array}{c} i=1\\ {\bar{x}}_{i}\in c_{k} \end{array}}^{n}{\left|S_{ij}\left(a,b_{i}\right)\right|}^{2}}$$
These dimensionless weights can easily locate any energy concentration or maximum values of correlation in the spectrogram, which, in turn, helps to extract the underline modal frequencies present in the measured response. This is due to the fact that the clusters, in this case, are, in principle, the energy localization or maximum correlations around the modes. Thus, the median of each cluster is identified as the frequency corresponding to that mode. Here, it is to be noted that apart from modal frequencies, there may be clusters corresponding to spurious frequencies, which need to be screened out from the pool. This is done by analyzing the phase portrait as discussed in the previous subsection. The optimum number of clusters is estimated from the gap statistics, where the gap value has the following form
(46)$$GV_{n}\left(k\right)=E_{n}\left\{\log W_{k}\right\}-\log\left(W_{k}\right)$$
In the above equation, $W_{k}$ represents the pool within the cluster, which is evaluated as
(47)$$W_{k}=\sum_{k=1}^{K}\frac{1}{2n_{k}}d_{k}$$
Here, the number of data point in the $k^{th}$ cluster is represented by $n_{k}$ whereas, $d_{k}$ is the sum of the pairwise distances for all points in that cluster. The optimal cluster number corresponds to the maximum or converged gap value.

### 3.4. Error Analysis

Figure 1 shows the schematic diagram of the algorithm used in this study to identify the random eigenvalues obtained from an ensemble of experiments. As the present work deals with the automated identification of random eigenvalues, it is worth studying the *pdf* that describes these random quantities and the error associated with the proposed identification process. The *pdf* describing the modal parameters can be obtained from their histogram. In this study, numerical *pdf*s are developed, which can be given an appropriate mathematical form by fitting suitable distributions using standard statistical tools, e.g., $\chi^{2}$ test or KS test [58]. Thus, the numerical evaluation of each outcome (i.e., eigenfrequencies and other parameters) in every test finally provides the probability distribution of the underlying random structural system.

Here, it may be noted that error estimation is difficult for field problems, where the actual values of the structural parameters are not known beforehand. However, as the present study uses a controlled laboratory experiment (where input parameters are known in every test), error estimation can be done to verify the accuracy of the proposed algorithm. It is described below to evaluate the reliability of the proposed identification strategy. The error function for each modal frequency is estimated using the experimental and identified values of the modal parameters. Thus, errors associated with the identification of modal frequencies can be expressed as
(48)$$\epsilon_{n}^{s}=\frac{\omega_{n}^{exp}-\omega_{n}^{ind}}{\omega_{n}^{exp}}\times 100\%$$
In the above expression, superscripts exp and ind correspond to the experimental and identified values of the modal frequencies, respectively. For a single sensor, the *pdf* of the error function can be evaluated considering all the test results, which is given by
(49)$$p\left(\epsilon^{*}\right)=\lim_{\Delta \epsilon \to 0}\frac{Prob\left[\epsilon <\epsilon (\bar{x})\leq \epsilon +\Delta \epsilon \right]}{\Delta \epsilon }$$
From the above *pdf*, the cumulative distribution function (*cdf*) can be estimated as follows
(50)$$P\left(\epsilon^{*}\right)=\int_{-\infty }^{\epsilon^{*}}p\left(\xi \right)d\xi$$
The significance of $P(\epsilon^{*})$ lies in its ability to offer the probability of detection of a particular mode from a specific sensor corresponding to an acceptable level of error $\epsilon^{*}$, which is discussed in details using numerical results in the following section.

## 4. Validation with Synthetic Experiment: 2DOF System

In this section, numerical examples are presented to show the performance of the proposed automated data-driven identification of random eigenvalues. First, the algorithm is validated using a 2-dof system [31], whose stiffness varies randomly. This simulated experiment helps to demonstrate the efficiency and accuracy of the proposed identification strategy. The subjected system has two masses of 1 kG and 1.5 kg, which are connected by 3 springs as shown in Figure 2a. The spring stiffness values, i.e., $k_{1}$, $k_{2}$ and $k_{3}$ are 1000 N/m, 1100 N/m and 100 N/m, respectively. Among these three springs, two have random stiffness parameters, which are modeled by the following expressions
(51)$$\begin{array}{ccc} k_{1} & = & \bar{k_{1}}\left(1+\varepsilon x_{1}\right)\\ k_{2} & = & \bar{k_{2}}\left(1+\varepsilon x_{2}\right) \end{array}$$
In the above expressions, $x_{1}$ and $x_{2}$ follow standard normal distribution. To ensure the positive definiteness of the stiffness matrix, a scaling parameter ε is considered, whose value is 0.25. The damping of this system is assumed to be 2% in all modes. The system is solved in state-space using a 4th order Runge–Kutta algorithm in MATLAB for free vibration with a unit initial velocity. Figure 2b shows a sample response of this system. The process is repeated to generate 15,000 samples of responses. Using this ensemble, proposed synchrosqueezed spectrum-based clustering is invoked to characterize its eigenvalues, i.e., modal frequencies and other associated parameters.

As the main objective of this study is to identify the *pdf* that dictates the nature of modal parameter from the measured acceleration response, wavelet-based time-frequency analysis is invoked as described in Section 3.1. In the preceding study [59], a comparison examination of the basic function of wavelet is conducted. Based on that result, complex Morlet wavelet is used with 732 scales covering the frequency range of 0.1 Hz to 10 Hz. The scalogram obtained for two *pdf*s from this step are shown in Figure 3a,c. These two figures clearly show two regions in each of them, where the signal energies are localized. However, it is difficult to pinpoint the scales corresponding to modal frequencies from these figures. This issue is resolved by the synchrosqueezed transform, which operates over the wavelet coefficients. It is invoked with Δω=0.01 rad/s and the scalogram for the two *dof*s are shown in Figure 3b and Figure 3d, respectively. These two figures clearly show the advantage of the reallocation/reassignment scheme adopted in synchrosqueezing, which distinctly identifies the scales corresponding to modal frequencies and helps to automate the process for ensemble-based data-driven identification for random systems. These figures also reveal that two modal frequencies are dominant in the response of 1st *dof* while only one mode is dominant in the 2nd *dof*. This is due to the fact that 2nd *dof* is connected to a stiffer spring that tries to prevent vibration compared to others and hence the 2nd mode is weaker in this signal. However, even this weaker mode is also identified by the clustering algorithm in the next step of the automation. For this purpose, the normalized energy spectrum is constructed following the expression in Equation (Equation 35). Using this scale-dependent energy, k-means clustering is adopted as discussed in Section 3.3. To do so, the optimal cluster number is estimated first with the help of gap statics as explained in Equations (Equation 46) and (Equation 47), which is shown in Figure 4a. From this figure, it can be observed that the optimal number of clusters in this example is 2 corresponding to the max gap value. Using this information, the normalized energies are clustered and the respective weights are estimated by Equation (Equation 45), which are shown in Figure 4b. In this figure, the values in x-axis are the median of two clusters, which are identified as the modal frequencies of the two *dof* systems. To verify it further, phase portraits corresponding to these two scales are studied further and their PSI values are checked. Figure 5 shows the phases obtained from inverse synchrosqueezed transform corresponding to the identified modal frequencies, which are in unison, i.e., reaching maxima or minima or zero-crossing together, indicating modal vibration. Corresponding PSI values obtained from Equation (Equation 38) for these two signals are 1.0, which confirms the identified frequencies as the modal frequencies.

Next, this process is repeated for the complete ensemble (i.e., 15,000 simulated responses) to estimate the underlying *pdf*s of the two modal frequencies. Figure 6a,c shows the numerical *pdf*s obtained from two *dof*s using simulated ensemble responses and are compared with their theoretical *pdf*s obtained using asymptotic integrals as described in Section 2.1 and Section 2.2. Figure 6a shows a close match between the exact and estimated values of the *pdf*s associated with these modal frequencies obtained from the measurement in 1st channel. Corresponding *cdf* of error is shown in Figure 6b, where the vertical line represents 5% error. This error estimate reveals that the reliability of the proposed automated data-driven uncertainty quantification of the random modal frequencies is more than 97.5%. Figure 6c shows the same *pdf*s obtained from the 2nd channel, whose error estimation is shown in Figure 6d. These two figures reveal that the 1st frequency is satisfactorily identified with a probability greater than 99%, while the 2nd mode shows higher error. The error in the 2nd mode is attributed to the weak energy content of the response in this mode in channel 2, as observed earlier in the wavelet spectrogram.

Once the frequencies are identified, attention is focused on the mode-shape estimation and subsequent quantification of the underlying uncertainties. As described in Equation (Equation 39), the mode-shape can be evaluated as the ratio of the two synchrosqueezed signals. However, the present study recommends the evaluation of mode shapes using model updating, which has certain advantages as described in the previous section. In this process, the objective function for model updating is formed based on the identified modal frequencies as described in Equation (Equation 40). Figure 7a,b shows the identified mode-shapes in light grey color, where the estimated mean modes-shapes are compared with theoretical values as described in Section 2. The *pdf*s in two modes are then estimated, which are shown in Figure 7c. It shows a close match indicating the efficiency of the proposed identification strategy, which is further demonstrated in terms of estimation error CDF in Figure 7d. This plot clearly shows that the probability of detection of these two mode-shapes is more than 99%.

Finally, an effort is made to estimate the damping ratio in the two modes. For this purpose, synchrosqueezed coefficients are used as described in Section 3.1. Here, it may be noted that the damping ratio in these two modes is assumed to be 2%, i.e., deterministic. However, the damping in the system is random as it is influenced by the modal masses and frequencies. Figure 8 shows the *pdf*s of the identified modal damping ratio obtained from two different channels. Though the damping ratio is constant in these modes, a variation is observed in the identification of these parameters due to the random nature of the stiffness properties and the error in estimation. However, the median values of the identified damping in these modes are 2.57% and 2.38%, respectively, which are very close to the theoretical value.

Overall, the performance of the automated identification of the modal parameters in light of system uncertainties is satisfactory, which is tested further using the experimental results.

## 5. Experimental Verification: Beam with Random Parameter

In this section, the methodology is implemented for a laboratory experiment on a beam model, whose mass changes randomly [60]. In this exercise, a flexible beam is set up with frequencies over a wider range, which helps establish the proposed strategy’s potential for similar problems encountered in physics and engineering.

In this example, a steel beam is considered whose length, width and depth are 1.2 m, 40.06 mm and 2.05 mm, respectively. The experimental setup is shown in Figure 9. On this beam, 12 movable masses, each having 2 gm weight, are attached with the help of a magnet. During each test, these movable masses are randomly placed to simulate an uncertain mass matrix. The test is repeated 100 times with different orientations of the movable masses. This beam is excited by a shaker with an impulse force at 50 cm from the left end. In each test, the shaker with the model number LDS V201 and serial number 92358.3 exerts a unitary force on the beam, and the reactions are recorded by three accelerometers, as shown in Figure 9. All three accelerometers are PCB type and the serial numbers are PCB 333M07 SN 25948, PCB 333M07 SN 26018 and PCB 333M07 SN 25942. A force transducer (series number: PCB 208C03 21487) is placed between the beam and the shaker to monitor the exerted force. Figure 10 shows a sample time history recorded with a 16,384 Hz sampling rate.

These recorded time histories are analyzed using a synchrosqueezed transformation with a complex Morlet basis function as in the previous example. The scalogram obtained from the wavelet transformation is shown in Figure 10b while its synchrosqueezed version is shown in Figure 10c. As obvious, the second scalogram has improved resolution showing the presence of different modal frequencies, which are identified by k-means clustering. A stability diagram is evaluated from test 1 data for comparative purposes, as illustrated in Figure 11. From this result, it is evident that it is unable to localize the lower frequencies and there are plenty of closely spaced frequencies on the higher side. As discussed by a previous researcher [8], it requires a number of screening methods to separate the spurious modes, and mathematical poles from this result to obtain an efficient conclusion. Figure 12 shows the clusters obtained from a sample response recorded using three different channels, which show 14 different clusters of energy localization after their gap statistics analysis. The median values of these clusters are identified as the modal frequencies of the beam. The process is repeated for the complete ensemble of tests and the identified frequencies are used for *pdf* estimation, which is shown in Figure 13 and Figure 14. These figures reveal that the proposed automated identification strategy could identify all the frequencies successfully. An interesting pattern of identified frequencies has emerged from these *pdf*s.

Except for the first mode, all other modes have been identified accurately. The first mode is very weak in this example, which was also reported by Adhikari and Phani [34]. Hence, the modeling in this reference and identification in the present work have encountered difficulties. Besides the first mode, the proposed automated data-driven strategy can efficiently identify all other modes with a high level of accuracy, as shown in the error *pdf*s in Figure 15. However, as observed in the previous example, estimated errors in a few frequencies are more in a particular channel, which is satisfactorily identified from other channels. For example, Figure 15 reveals that mode 7 has more than 99% chance of detection from channels 1 and 2, while the same frequency faces 90% rate of detection in channel 3. However, these are only a few cases (e.g., mode 10, 12) that show this pattern and are envisaged due to the relative location of the accelerometer in the light of the deformed shape of the beam in that mode. The phase portraits of the inverse synchrosqueezed signals are then analyzed to ensure that these clusters correspond to modal frequencies. Figure 16 demonstrates two randomly selected phase portraits corresponding to 2nd and 5th modes. The three-phase angles are clearly in unison as their portraits show linear patterns with a PSI value of 1. All these results suggest that the proposed data-driven automated identification procedure can successfully identify all the modes along with their underlying *pdf*s for uncertainty quantification. In this context, results corresponding to mode-shape and damping are not presented in this example as their actual values are not readily available for comparison. However, the proposed methodology does not face any difficulty in identifying them as was demonstrated in the previous example.

Based on the *pdf*s of 14 modal frequencies of this experimental validation problem, it can be inferred that the proposed automated data-driven scheme offers satisfactory performance for uncertainty quantification associated with random eigenvalue problems.

## 6. Conclusions

An inverse value problem is described above with uncertainty in the system itself. For this purpose, synchrosqueezed transformation is coupled with k-means clustering for output-only quantification of modal parameters in the statistical sense, which is the major contribution of this study. This automated process is repeated for the ensemble of experiments to quantify the underlying randomness associated with the modal parameters. The following conclusions are drawn based on the numerical results presented in the previous section

Overall, the proposed data-driven identification scheme can successfully track the eigenvalues of an uncertain linear system from the measured responses with sufficient accuracy. The automated methodology is primarily based on the synchrosqueezed transform, which offers an improved resolution of the energy and phase spectra, which are analyzed with the help of k-means clustering for mode localization and subsequent parameter estimation. This entire process does not require either any prior knowledge of the system and/or input nor any user intervention for reverse analysis, which is clearly an advantage compared to other similar methods available in the literature.The error analysis establishes the precision offered by the proposed identification scheme, as the *pdf*s of different modal parameters closely match their theoretical models. The error CDF also shows the probability of successful detection of a particular mode corresponding to the acceptable error limit. For example, the probability of detection is greater than 99% in most cases, corresponds to an error of 5%, acceptable for all practical purposes. The error value mentioned here is composed of two errors: detection and system uncertainty. It is extremely rare to distinguish between these two types of errors in practical application. It is also observed that the success rate of detecting a few frequencies from a particular sensor is poor; however, the same frequency is successfully identified with a high level of accuracy from another sensor. Thus, the study demonstrates that sensor locations affect the quality of the end results, which, in turn, advocates deciding the optimal sensor location in a probabilistic sense.Unlike many other learning-based algorithms, the proposed methodology requires no training data or pre-tuning for the model. For this reason, the proposed automated data-driven identification scheme has a broader scope for the inverse parameter estimation of a large class of systems in different sciences and engineering disciplines. It has a wider scope for alternative parameter estimation. Due to being able to detect small changes in modal parameters, it is inherently capable of detecting subtle changes, allowing it to be used to identify damages and control vibrations based on feedback within structures.

## Figures and Tables

**Figure 1 sensors-23-03421-f001:**
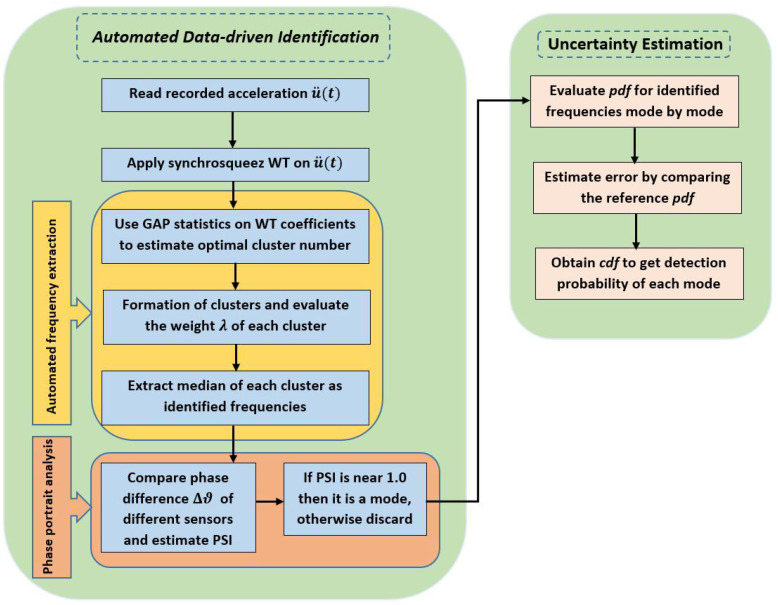
Proposed automated data-driven methodology.

**Figure 2 sensors-23-03421-f002:**
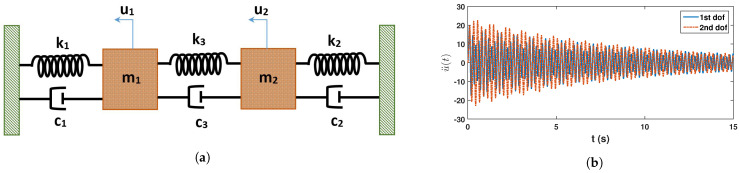
2 dof system; (**a**) structural details & (**b**) sample acceleration response.

**Figure 3 sensors-23-03421-f003:**
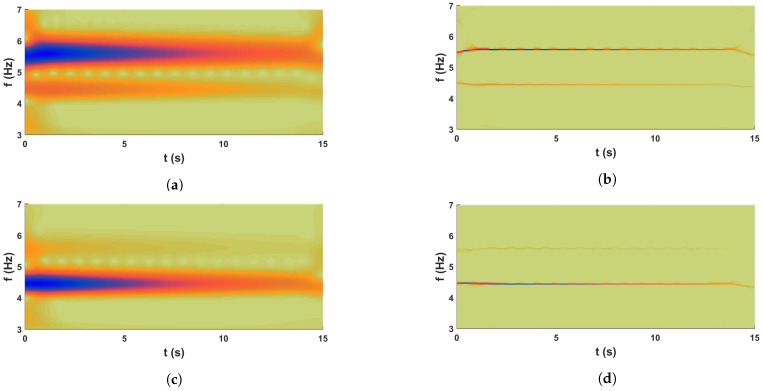
Scalogram; (**a**,**c**) are wavelet transform of $u_{1}$ & $u_{2}$, respectively, and (**b**,**d**) are synchrosqueezed transform of $u_{1}$ & $u_{2}$, respectively.

**Figure 4 sensors-23-03421-f004:**
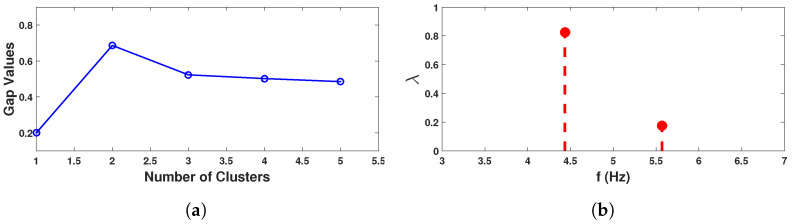
Clustering; (**a**) Gap statistics for optimal cluster & (**b**) Weights of each cluster.

**Figure 5 sensors-23-03421-f005:**
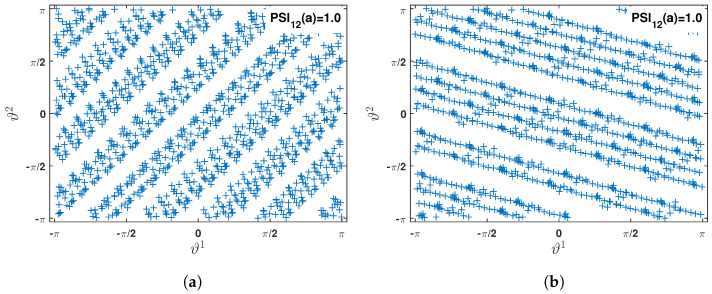
Comparison of modal responses and respective phases: (**a**) mode 1 and (**b**) mode 2.

**Figure 6 sensors-23-03421-f006:**
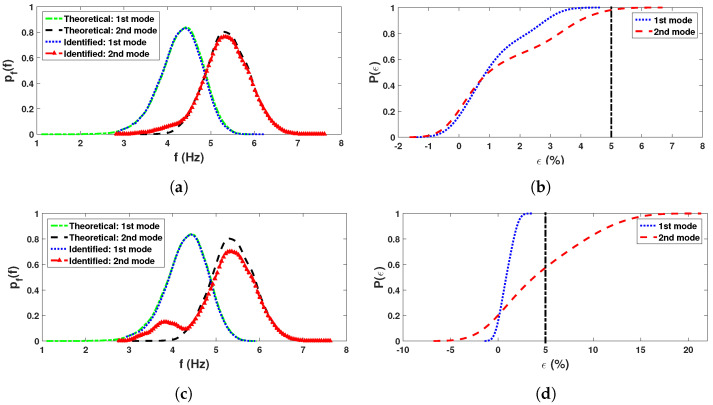
Comparison between theoretical and identified: (**a**) identified *pdf*s & (**b**) *cdf*s of error from 1st *dof* response; (**c**) identified *pdf*s & (**d**) *cdf*s of error from 2nd *dof* response.

**Figure 7 sensors-23-03421-f007:**
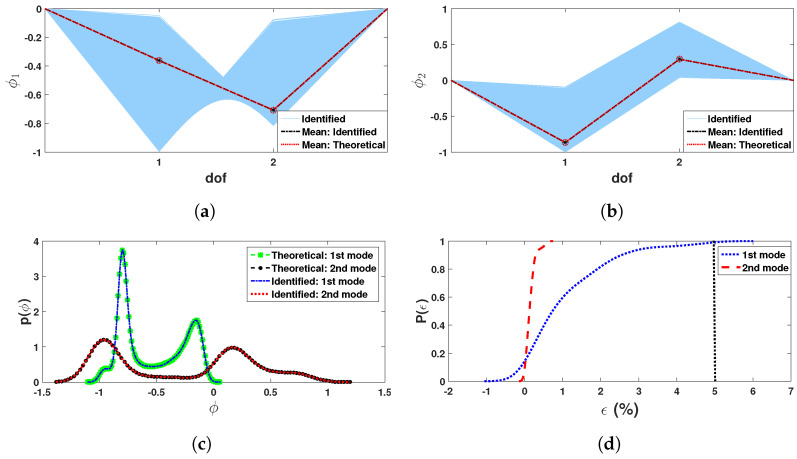
Modeshape estimation: identified (**a**) modeshape 1 & (**b**) modeshape 2; (**c**) *pdf*s of modeshape and (**d**) *cdf*s of error in modeshape estimation.

**Figure 8 sensors-23-03421-f008:**
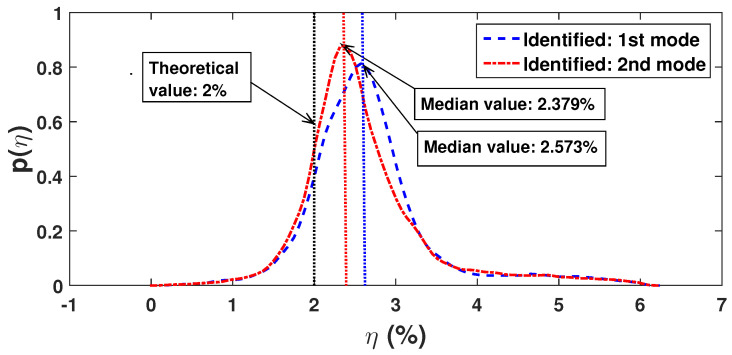
Identified modal damping.

**Figure 9 sensors-23-03421-f009:**
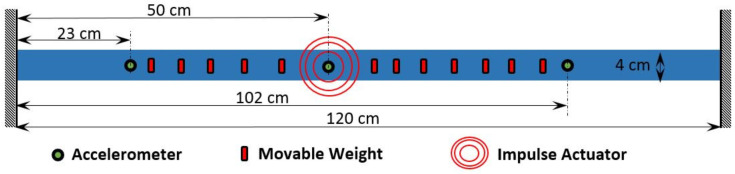
Schematic diagram of experimental setup [60].

**Figure 10 sensors-23-03421-f010:**
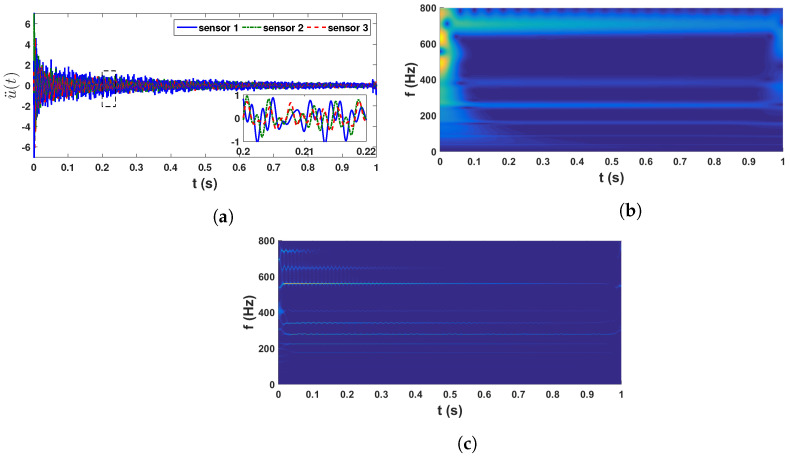
Recorded beam response in test 1 and Scalogram; (**a**) accelerations in three channels; (**b**) wavelet scalogram of sensor 2 & (**c**) synchrosqueezed scalogram of sensor 2.

**Figure 11 sensors-23-03421-f011:**
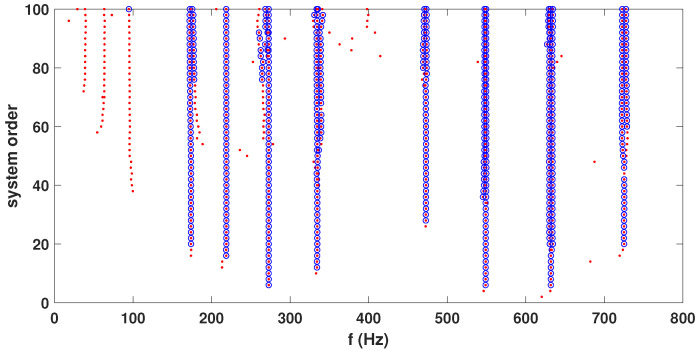
Stabilization diagram of test 1 of all three channels.

**Figure 12 sensors-23-03421-f012:**
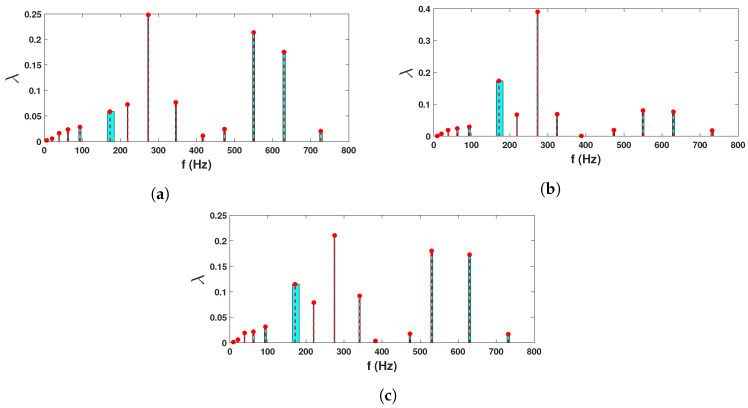
Clustering of beam test data from different sensors; (**a**) sensor 1, (**b**) sensor 2 & (**c**) sensor 3.

**Figure 13 sensors-23-03421-f013:**
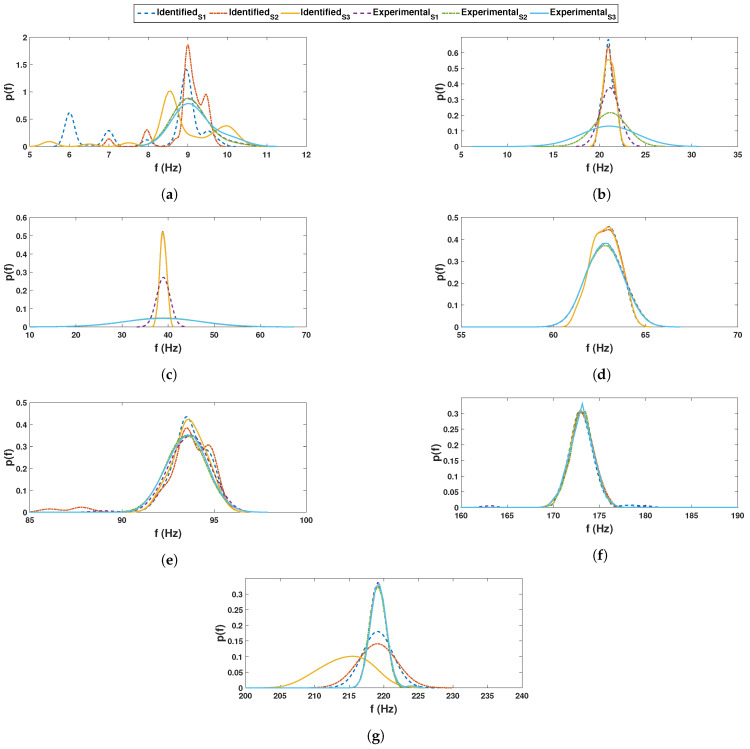
Comparison of *pdf* of identified and experimental cases for the first 7 frequencies for beam experiment.

**Figure 14 sensors-23-03421-f014:**
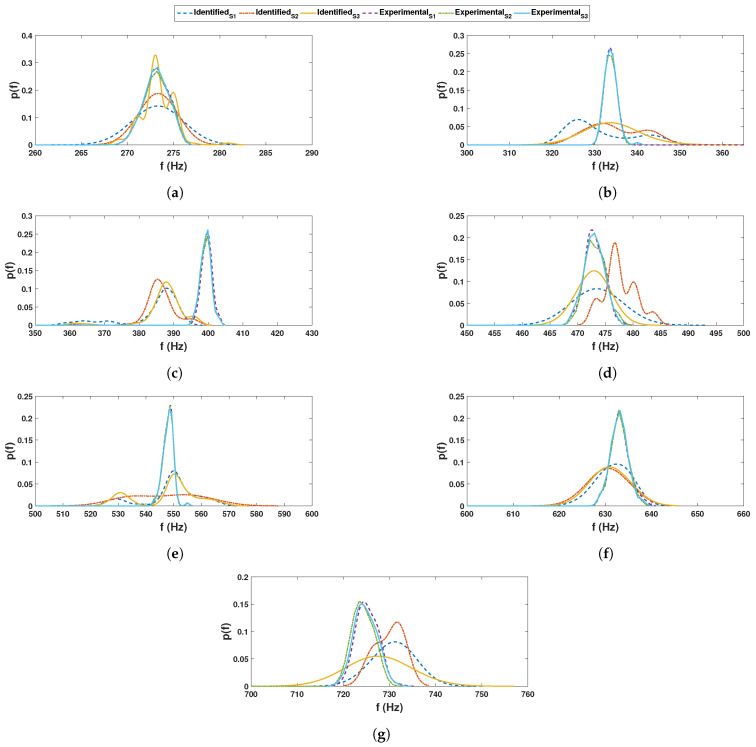
Comparison of *pdf* of identified and experimental cases for last 7 frequencies for beam experiment.

**Figure 15 sensors-23-03421-f015:**
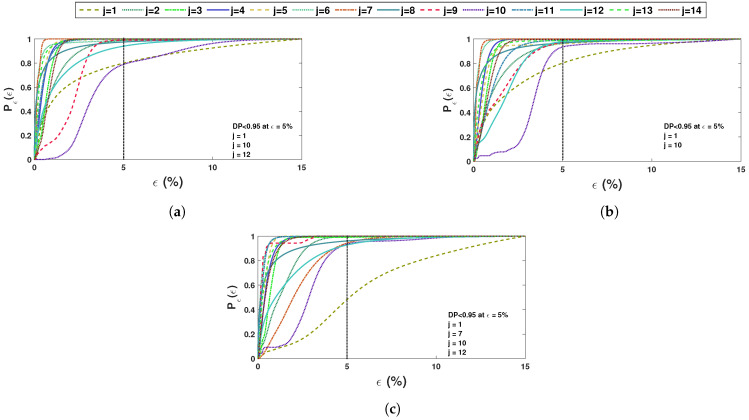
Detection probability (DP) of beam frequencies for different sensors (**a**) 1st, (**b**) 2nd and (**c**) 3rd sensor [NB: DP represents ’detection probability’ of a particular mode].

**Figure 16 sensors-23-03421-f016:**
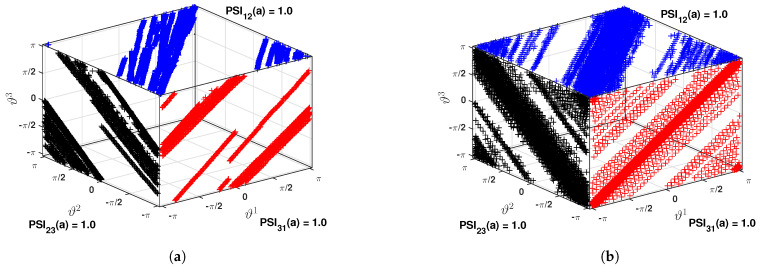
Phase and PSI from different channels: (**a**) mode 2 and (**b**) mode 5.

## Data Availability

Not applicable.

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
