# Peer review of "A Data-Driven System Identification Method for Random Eigenvalue Problem Using Synchrosqueezed Energy and Phase Portrait Analysis"

_sensors, 2023, doi:10.3390/s23073421_

Round 1

Reviewer 1 Report

1. English proofreading is suggested to improve the English Language.

2. Please state the previous methods applied, and the flaws in these methods.

3. On what basis was the complex morlex function selected?

4. Align the labels of Figures 10 to 15.

5. Which sensor reading was applied in analysis provided in Section 4.2?

6. Line 558 should be Fig 10b and Fig 10c, not Fig 10a and Fig 10b.

7. Can the authors explain why the first mode was undetected while higher modes were detected. It is known that the first mode is the easiest to estimate.

8. The following research are related to the topic of study, I suggest benefiting from and citing them:

  • https://doi.org/10.1016/j.jsv.2019.115069
  • https://doi.org/10.3390/app11020770
  • https://doi.org/10.1016/j.measurement.2021.109464
  • https://doi.org/10.1016/j.measurement.2022.112160

Reviewer 2 Report

The authors work is original, interesting and well presented. The findings and conclusions are consistent with the overall purpose of the paper. However, the study is limited to a simple numerical case and a simple experimental one. The authors should expand with a more complicated study case. The only comments that I have are the following

The references are pretty out to date. The authors should have a more recent discussion about the automated system ID techniques currently available in the field. There are plenty related to SSI. Here some of the most recent works

[1] Mugnaini, Vezio, Luca Zanotti Fragonara, and Marco Civera. "A machine learning approach for automatic operational modal analysis." Mechanical Systems and Signal Processing 170 (2022): 108813.

[2] P. E. Charbonnel, Fuzzy-driven strategy for fully automated modal analysis: Application to the smart2013 shaking-table test campaign, Mechanical Systems and Signal Processing 152(2021) 107388.[3] E. Tronci, M. De Angelis, R. Betti, and V. Altomare, "Multi-stage semi-automated methodology for modal parameters estimation adopting parametric system identification algorithms," Mechanical Systems and Signal Processing, vol. 165, p. 108317, 2022.

[4] E. Neu, F. Janser, A. A. Khatibi, A. C. Orifici, Fully automated operational modal analysis using multi-stage clustering, Mechanical Systems and Signal Processing 84 (2017) 308-323.

The quality of Figure 1 should be improved.

The application of the methodology to a very simple numerical case and another simple experimental study is highly limiting. There are plenty of open access dataset the authors could access to in order to show the applicability of the methodology in those cases.

Reviewer 3 Report

1. The author needs to further refine the Abstract。

2. The author can try to delete part of the content of the paper appropriately to make the intention of the paper more prominent, especially Section 2 and 3.

3. The numerical simulation and experimental results are clear, and the paper may be fuller if some comparative analysis is added.

4. Some of the references are too old, so it is suggested that the author refer to more recent references.
